# Taxifolin and Lucidin as Potential E6 Protein Inhibitors: p53 Function Re-Establishment and Apoptosis Induction in Cervical Cancer Cells

**DOI:** 10.3390/cancers14122834

**Published:** 2022-06-08

**Authors:** Diana Gomes, Shivani Yaduvanshi, Samuel Silvestre, Ana Paula Duarte, Adriana O. Santos, Christiane P. Soares, Veerendra Kumar, Luís Passarinha, Ângela Sousa

**Affiliations:** 1CICS-UBI—Health Sciences Research Centre, University of Beira Interior, Avenida Infante D. Henrique, 6200-506 Covilhã, Portugal; diana.gomes@ubi.pt (D.G.); sms@ubi.pt (S.S.); apcd@ubi.pt (A.P.D.); aos@ubi.pt (A.O.S.); lpassarinha@fcsaude.ubi.pt (L.P.); 2Associate Laboratory i4HB—Institute for Health and Bioeconomy, Faculdade de Ciências e Tecnologia, Universidade NOVA, 2819-516 Caparica, Portugal; 3UCIBIO—Applied Molecular Biosciences Unit, Departamento de Química, Faculdade de Ciências e Tecnologia, Universidade NOVA de Lisboa, 2829-516 Caparica, Portugal; 4Amity Institute of Molecular Medicine and Stem Cell Research (AIMMSCR), Amity University, Noida Uttar Pradesh 201303, India; shivani.yaduvanshi1@s.amity.edu; 5Laboratório de Fármaco-Toxicologia-UBIMedical, Universidade da Beira Interior, 6200-284 Covilhã, Portugal; 6CNC—Center for Neuroscience and Cell Biology, University of Coimbra, 3004-504 Coimbra, Portugal; 7C4—Cloud Computing Competence Centre, UBIMedical, University of Beira Interior, Estrada Municipal 506, 6200-284 Covilhã, Portugal; 8Department of Clinical Analysis, School of Pharmaceutical Sciences, São Paulo State University (UNESP), Campus Ville, Araraquara, São Paulo 14800-903, Brazil; soarescp@fcfar.unesp.br

**Keywords:** cervical cancer, E6 protein inhibitors, human papillomavirus, in silico tools, lucidin, p53, taxifolin, molecular docking

## Abstract

**Simple Summary:**

Human papillomavirus (HPV)-related cancers continue to be a major medical concern, and there exists an urgent need to improve the current therapeutic approaches by combining strategies or proposing new compounds to offer more specific and less invasive treatments. The aim of this work was to discover potential inhibitors of the E6/E6AP/p53 complex formation. We started this work with an initial in silico approach including molecular docking and molecular dynamics simulations, and these tools allowed us to select potential inhibitors, using E6 protein as a target. In addition, we found that lucidin and taxifolin were able to selectively decrease the viability of HPV-positive cells to re-establish p53 protein levels and to induce apoptosis. These findings represent a promising starting point for the development of anti-HPV drugs.

**Abstract:**

Cervical cancer is the fourth leading cause of death in women worldwide, with 99% of cases associated with a human papillomavirus (HPV) infection. Given that HPV prophylactic vaccines do not exert a therapeutic effect in individuals previously infected, have low coverage of all HPV types, and have poor accessibility in developing countries, it is unlikely that HPV-associated cancers will be eradicated in the coming years. Therefore, there is an emerging need for the development of anti-HPV drugs. Considering HPV E6’s oncogenic role, this protein has been proposed as a relevant target for cancer treatment. In the present work, we employed in silico tools to discover potential E6 inhibitors, as well as biochemical and cellular assays to understand the action of selected compounds in HPV-positive cells (Caski and HeLa) vs. HPV-negative (C33A) and non-carcinogenic (NHEK) cell lines. In fact, by molecular docking and molecular dynamics simulations, we found three phenolic compounds able to dock in the E6AP binding pocket of the E6 protein. In particular, lucidin and taxifolin were able to inhibit E6-mediated p53 degradation, selectively reduce the viability, and induce apoptosis in HPV-positive cells. Altogether, our data can be relevant for discovering promising leads for the development of specific anti-HPV drugs.

## 1. Introduction

Gynecologic malignancies correspond to any cancer that starts in the female reproductive system and includes cervical, ovarian, uterine, vaginal, and vulvar cancers. According to GLOBOCAN 2020, cancer prevalence and mortality are rapidly rising worldwide, where cervical cancer (CC) has the highest incidence and death among all gynecological cancers [1]. In particular, the CC incidence is more notorious in Africa, Asia, and South America (regions in development), which could be due to limited resources, lack of screening programs, and access to health care [2,3]. Cervical cancer is associated with the establishment of a persistent infection by human papillomavirus (HPV), where HPV16 and 18 remain the major contributors. The oncogenic role of HPV is highly related to the activity of two viral oncoproteins, E6 and E7, that disrupt p53 and pRb tumor suppressor pathways, respectively, resulting in the immortalization and cellular transformation of infected cells [4,5]. Despite the fact that preventive anti-HPV vaccines are available, they have low coverage for all HPV types and do not exert a therapeutic effect in individuals previously infected [6]. In addition, the current treatments for CC consist of surgery, radiotherapy, and chemotherapy, which affect diseased and healthy tissues alike, causing several side effects, and have a high cost [7]. Therefore, there is an urgent need for a specific, efficient, and non-invasive therapeutic strategy for CC management.

The HPV16 E6 protein is composed of 158 amino acids and 2 zinc-binding motifs, and its most important function is escaping cell death by the induction of p53 degradation. In fact, E6 interacts with the LxxLL motif of the E6-associated protein (E6AP), an E3 ubiquitin ligase, inducing p53 degradation through the proteasome pathway and, consequently, blocking p53 dependent apoptosis [8]. Although E6AP is not directly involved with the E6/p53 interaction, it is absolutely required to mediate this interaction due to the conformational changes and stability induced in the E6 protein. Besides structuring the p53 binding cleft on E6, the LxxLL binding pocket is crucial for the tumorigenic phenotype of E6 proteins [9]. 

Recently, some researchers have been trying to propose new strategies for the inhibition or blockage of E6 oncoprotein as anticancer therapies by using interference RNA (16E7-shRNA), inhibitory peptides (pep11**), or potential small molecule drugs (e.g., luteolin) [10,11,12,13]. Given that the E6/E6AP crystal structure [9] and the trimeric complex E6/E6AP/p53 crystal structure are available on the protein data bank (PDB) [14], it is possible to understand the amino acid (aa) residues involved in different interactions with the E6 protein. In particular, the residues that form the LxxLL-binding pocket are the most interesting, since this site represents a promising target for the discovery of therapeutic drugs. In line with this, computer-aided drug design techniques have been employed to reduce time and costs as well as accelerate drug development [15,16]. For example, methods such as molecular docking and molecular dynamics simulations have been used to select promising molecules from large libraries by predicting the orientation of a compound to the target, their interactions, and binding stability [17]. 

Natural compounds have gained major attention as anticancer agents, being easily accessible and affordable, and usually present less toxicity when compared to other molecules. A relevant example are flavonoids, which have been shown to have antibacterial, antiviral, and antioxidant properties and are able to induce apoptosis in cancer cells [18]. 

In the present work, we performed an in silico screening using the E6/E6AP/p53 trimeric complex structure (PDB ID: 4XR8, consulted on 5 February 2021) and different isolated natural compounds to search for potential E6 inhibitors. We identified two compounds, lucidin and taxifolin, able to increment the level of the p53 tumor suppressor protein and, consequently, induce apoptosis in HPV-positive cancer cells. The results reported could be a starting point for the design and the development of new anti-HPV drug candidates.

## 2. Materials and Methods

### 2.1. Materials

DMEM/F12, DMEM high glucose, and RPMI 1640 cell culture media were purchased from Gibco (Thermo Fisher Scientific, Waltham, MA, USA). 3-(4,5-dimethylthiazol-2-yl)-2,5-diphenyltetrazolium bromide (MTT) was obtained from Alfa Aesar (Waltham, MA, USA). Luteolin and lucidin were purchased from PhytoLab (Vestenbergsgreuth, Germany), and alizarin and taxifolin were acquired from Sigma Aldrich (St. Louis, MO, USA). Propidium iodide solution and Hoechst 33,342 were also bought from Sigma Aldrich (St. Louis, MO, USA). A Caspase 3/7 Glo Kit was obtained from Promega (Madison, WI, USA). All the other chemicals and solvents obtained commercially were of analytical grade. All solutions were freshly prepared by using ultra-pure grade water purified with a Milli-Q system from Millipore (Billerica, MA, USA). 

### 2.2. Methods

#### 2.2.1. In Silico Studies

##### E6 Protein Preparation 

The three-dimensional structure of the HV16 E6 protein (PDB code: 4XR8) was obtained from the protein data bank (PDB) [14]. The E6 protein (Chain F) and water molecules were selected using the software Chimera (v. 1.10.1), and the final structure was stored in PDB format. Then, nonpolar hydrogen atoms were merged in AutoDockTools (v. 1.5.6), and Kollman and Gasteiger partial charges were added. Ultimately, the prepared structure was converted from the PDB format to PDBQT for subsequent use in docking studies.

##### Ligands Preparation

All ligands were created using ChemDraw (v. 12.0) software (by Cambridge ChemBioOffice 2010). Energy minimization and geometry optimization were performed with Chem3D (v. 12.0), and the final structures were saved in PDB file format. Then, the ligands were fully prepared by choosing torsions, and the structures were converted from PDB format to PDBQT in AutoDockTools software.

##### Grid Map Calculations 

Grid parameters were chosen using AutoGrid4, based on the E6AP binding site coordinates of the E6 protein crystal structure. The size of the grid box was 50 × 50 × 50 with 0.375 Å of spacing. 

##### Molecular Docking Simulations

Molecular docking simulations were conducted using the Lamarckian genetic algorithm and empirical free energy scoring function. The maximum number of energy evaluations was 2,500,000 and the GA population size was 150. A total of 15 hybrid global-local Lamarckian search (GA-LS) runs were performed for each simulation. The results of molecular docking were visualized in the Biovia Discovery studio visualizer (Dassault Systèmes) and PyMol program (PyMol Molecular Graphics System, v. 1.3, Schrodinger, LLC e www.pymol.org (accessed on 3 March 2021)), built for educational use. 

##### Molecular Dynamics Simulations

Molecular dynamics (MD) simulations were performed for protein E6-alizarin/taxifolin/lucidin docked complexes. All simulations were performed using Gromacs 2020.2 for 200 nanoseconds (ns) using the CHARMM36 force field and tip3p water model [19,20]. Simulation inputs were built using the CHARMM-GUI web program [21,22]. Protein molecules were placed in a cubic box located 10 Å away from box boundaries. The simulation time was selected by observing the root-mean-square deviation (RMSD) profile. The systems were neutralized with 0.15 M NaCl. Simulations were performed with periodic boundary conditions. The energy minimization steps comprising gradual reduction of side-chain and backbone restraints were carried out for 250 picoseconds (ps). Water equilibration around the protein molecule was performed under NVT for 125 ps followed by 125 ps NPT ensembles at 303 K. The production run was performed at 303 K in the NPT ensemble. The time step was 2 femtoseconds (fs) and the trajectory was saved every 10 ps. The temperature was maintained using velocity scaling. Bond lengths were constrained with the LINCS algorithm. The pressure was controlled by isotropic coupling using the Parrinello–Rahman barostat. A Verlet scheme was used for van der Waals and Particle Mesh Ewald electrostatics (PME) interactions within 1.2 nm. Van der Waals interactions were switched above 1.0 nm. The progress of the simulations was monitored by RMSD profiles. After the production run, the MD results were analyzed in Gromacs 2020.2 package utilities. 

#### 2.2.2. In Vitro Studies

##### Cell Culture

NHEK cells are Primary Normal Human Epidermal Keratinocytes, while C33A, HeLa, and Caski cells are cancer cells (C33A are HPV negative, HeLa are HPV18 positive, and Caski are HPV16 positive cells). NHEK and HeLa were obtained from PromoCell (Heidelberg, Germany), C33A were kindly provided by MD PhD Aldo Venuti, IRCCS Regina Elena National Cancer Institute in Italy, and Caski cells were also generously offered by Professor José das Neves, University of Porto, Portugal. NHEK and HeLa were cultured in DMEM/Ham’s F-12 nutrient mixture (DMEM-F12), C33A were cultured in DMEM high glucose, and Caski were cultured in RPMI 1640 medium. All media were supplemented with 10% (*v*/*v*) fetal bovine serum and a mixture of penicillin (100 mg/mL) and streptomycin (100 mg/mL). Cells were maintained/incubated at 37 °C, 5% CO_2_, and 95% humidity, and media was renewed every 2–3 days until cells almost reached a confluence state. When cells reached approximately 80/90% confluence, they were gently detached by trypsinization (trypsin-EDTA solution containing 0.125 g/L of trypsin and 0.02 g/L of EDTA). Prior to every experiment, viable cells were counted in a Neubauer chamber by a trypan blue exclusion assay and diluted in the appropriate complete cell culture medium.

##### Preparation of Compound Solutions

All compounds (luteolin, lucidin, taxifolin, and alizarin) were dissolved in dimethyl sulfoxide (DMSO; Sigma-Aldrich, Inc., St. Louis, MO, USA) at 10 mM concentration and stored at 4 °C (protected from light). From the stock solutions, different concentrations were prepared in a complete culture medium before each experiment.

##### Cell Viability Assays

Cell viability was determined by the 3-(4,5-dimethylthiazol-2-yl)-2,5-diphenyl tetrazolium bromide (MTT) method. To test the compounds’ cytotoxicity in cell lines, 5 × 10^3^ HeLa, 1.5 × 10^4^ CaSki, 4 × 10^4^ C33A, and 5 × 10^3^ NHEK cells were seeded into 96-well plates and, the next day, treated with each compound (10 and 100 μM for preliminary studies and 0.1, 1, 10, 25, 50, 100, 150, and 250 μM for concentration-response studies) [12,23]. After 24 or 48 h, cells were washed with 100 μL of phosphate buffer saline (PBS; 137 mM NaCl, 2.7 mM KCl, 10 mM Na_2_HPO_4_ and 1.8 mM KH_2_PO_4_, pH 7.4), and then 100 μL of the MTT solution (0.5 mg/mL), prepared in the appropriate serum-free medium, was added to each well, followed by incubation for approximately 4 h at 37 °C. Then, the medium was removed, and formazan crystals were dissolved in DMSO. Absorbance was measured at 570 nm using the microplate reader Bio-rad xMark spectrophotometer. Ethanol-treated cells and luteolin were used as positive controls. Each experiment was performed with four replicates and independently repeated three times. Cell viability values were expressed as percentages relative to the absorbance determined in the cells used as negative controls (DMSO-treated cells).

##### Western Blot Analysis

HeLa and NHEK cells seeded at 1.5 × 10^5^ per well in 6-well plates were treated with test compounds for 48 h. The p53 and Bax protein expression in cells treated with the concentration of lucidin, taxifolin, and alizarin that causes cell viability to decrease by 50% was measured by Western blot analysis. C33A cells, seeded at 6 × 10^5^ per well in 6-well plates, were treated with test compounds for 48 h, and p53 and BAX protein levels were assessed. Whole-cell extracts were obtained with lysis buffer (25 mM base Tris, 2.5 mM EDTA, 1% Triton X-100, 2.5 mM phenylmethylsulfonyl fluoride (PMSF), and EDTA-free protease inhibitor cocktail (Roche)). Lysates were incubated on ice for 10 min and then centrifuged at 10,000× *g* for 1 min at 4 °C. Supernatants were recovered and protein concentrations measured using the Pierce BCA Protein Assay Kit (Thermo Scientific, Waltham, MA, USA). Equal amounts of protein (30 μg) from supernatants were denatured (100 °C, 5 min) and separated by sodium dodecyl sulfate-polyacrylamide gel electrophoresis (SDS–PAGE). Following this, proteins were transferred onto PVDF membranes (750 mA, 90 min) that were blocked for 1 h with 5% (*w*/*v*) nonfat dry milk in TBS-T (Tris-buffered saline containing 0.1% Tween 20). Membranes were incubated with anti-p53 primary antibody (1:200) (Santa Cruz Biotechnology, Dallas, TX, USA), anti-p21 primary antibody (1:200) (Santa Cruz Biotechnology, Dallas, TX, USA), and anti-BAX primary antibody (1:1000) (Cell Signaling Technology, Danvers, MA, USA), respectively, overnight at 4 °C, followed by treatment with anti-mouse secondary antibody (1:5000) (Santa Cruz Biotechnology, Dallas, TX, USA) and anti-rabbit secondary antibody (1:5000) (Sigma–Aldrich, St. Louis, MO, USA) at room temperature for 2 h. Lastly, membranes were incubated in β-actin primary antibody (1:20 000) (A3854, Sigma–Aldrich, St. Louis, MO, USA). ECL substrate (BioRad, Hercules, CA, USA) was used to signal detection, and images were acquired by using a ChemiDoc™ XRS system (BioRad, Hercules, CA, USA) and analyzed with Image Lab software (BioRad, Hercules, CA, USA).

##### RT-PCR Analysis

HeLa cells seeded at 6 × 10^4^ per well in 12-well plates were treated with test compounds for 24 h. The p53 mRNA transcripts in cells treated with the concentration of lucidin, taxifolin, and alizarin required for the reduction of cell viability by 50% were measured by RT-PCR. To extract total RNA, the cells were lysed through the addition of TRIzol (GriSP, Porto, Portugal) (250 μL) and incubated at room temperature for 5 min, followed by the addition of chloroform and vigorous stirring, following the manufacturer’s guidelines. The cDNA synthesis was generated from RNA (1 μg) and performed by using the Xpert cDNA Synthesis Kit from Grisp (GriSP, Porto, Portugal), following the manufacturer’s protocol. PCR amplification of p53 cDNA was performed by using the following primers: primer reverse (5′-CCT CAT TCA GCT CTC GGA AC-3′) and primer forward (5′-CCT CAC CAT CAT CAC ACT GG-3′). Samples were then placed in a T100™ Thermal Cycler (Bio-Rad Laboratories, Inc., Hercules, CA, USA) with the following conditions: 95 °C for 5 min, 29 cycles of 30 s at 95 °C, 30 s at 60 °C, 1 min at 72 °C, and finally, 10 min at 72 °C. PCR products were analyzed by electrophoresis on an agarose gel and visualized in a UVItec Gel documentation system under UV light (UVItec Limited, Cambridge, UK).

##### Caspase-3/7 Assay

The activity of caspase-3 and caspase-7 was measured using the Caspase-Glow^®^ 3/7 Assay (Promega, Madison, WI, USA), following the provided instructions. Briefly, HeLa cells were incubated with lucidin, taxifolin, and alizarin for 48 h. Incubation with 1 μM of staurosporine was used as a positive control during the same period. The activity of caspase-3/7 was measured through a homogeneous luminescent assay for 2 h. Each experiment was independently repeated three times with quadruplicate replicates. 

##### Hoechst/Propidium Iodide (PI) Staining

HeLa cells, seeded at 6 × 10^4^ per well in 12-well plates, were treated with test compounds for 48 h. Then, cells were collected and washed with PBS to a density of 5 × 10^5^ cells/mL. Hoechst 33,342 (10 ug/mL) was added to the cell suspension and incubated at 37 °C for 10 min. Thereafter, cells were centrifuged, resuspended in PBS, and PI (1 μg/mL) was added. After 15 min of incubation in the dark at room temperature, cells were placed on a u-slide 8-well chamber (Ibidi GmbH, Gräfelfing, Germany). The slides were immediately observed using an LSM 710 confocal laser scanning microscope (Carl Zeiss, Oberkochen, Germany) under 40× magnification and analyzed with the Zeiss LSM 710 laser scanning confocal microscope (Carl Zeiss SMT, Inc., Oberkochen, Germany). The percentage of apoptotic cells was estimated by counting the number of chromatin condensed-positive nuclei and Hoechst-stained nuclei in six randomly selected 40× magnification images. The ratio between the number of chromatin condensed-positive nuclei and the total number of intact nuclei was calculated [24,25]. 

##### Flow Cytometry 

A cell cycle analysis and quantification of sub-G1 events were performed by flow cytometry after staining with PI. Briefly, HeLa cells were seeded in 6-well plates at 1.5 × 10^5^ cells per well and after 24 h were treated with lucidin, taxifolin, and alizarin for 48 h or 72 h. Luteolin-treated cells were used as a positive control. Then, supernatants and cells were collected and resuspended in 500 μL of a PBS cold solution, followed by fixation with 70% of EtOH and kept at −20 °C. After 2 days, the fixed cells were washed with PBS, resuspended in a solution of disodium hydrogen phosphate containing 0.1% of Triton X-100, and incubated at room temperature for 5 min. Then, cells were resuspended in a solution of PI (20 μg/mL) and Ribonuclease A (10 μg/mL) in PBS, and incubated for 30 min in the dark. A minimum of 20,000 events were acquired using a BD FACSCalibur (San Jose, CA, USA) flow cytometer, and the analysis was performed with FlowJo software. 

##### Statistical Analysis

Data from at least three independent experiments were expressed as mean ± standard deviation. Statistical significance was defined as significant when the *p*-value ≤ 0.05. The dose–response curves were built using a sigmoidal dose–response (variable slope) curve fit with a 95% confidence interval. The compound concentration required for the reduction of cell viability by 50% was determined by interpolation at 95% confidence. Differences between groups were evaluated by employing the Student *t*-test and one-way analysis of variance with the Bonferroni test on GraphPad Prism v.8.01 (GraphPad Software Inc., San Diego, CA, USA).

## 3. Results

### 3.1. In Silico Screening and Natural Compounds Identification 

Several natural products, including flavonoids, have the ability to interact with proteins and modulate several biological events related to cancer, such as apoptosis, vascularization, cell differentiation, and cell proliferation, among others [26]. We used molecular docking to screen the potential of different isolated natural products, including flavonoids, to inhibit the E6/E6AP/p53 complex formation. The HPV16 E6 protein structure was extracted from the E6/E6AP/p53 ternary complex (PDB code: 4XR8) [14] to be used as the target, and the natural products presented in Table 1 were used as ligands in the E6AP binding site of the E6 protein, resorting to the Autodock Tools software. Luteolin, a flavonoid already described in the literature as capable of re-activating the p53-mediated pathway in HPV-positive cells [10,27] by impairing the binding of E6/E6AP, was used as a starting/comparison point for the molecular docking studies. In this way, we first estimated a score for the binding affinity of luteolin for the E6 protein. This site was defined considering the described amino acid residues (Cys51, Tyr70, Leu67, Gln107, Leu50, Tyr32, Val31, Val62, Arg55, Arg131) of the E6 hydrophobic pocket that mediate crucial contacts with E6AP [9], and also because it is considered a druggable binding pocket [11]. The obtained docking score was −6.28 Kcal/mol, and the molecular interactions of the docked complex included specific residues that are important for E6/E6AP interaction (Figure 1). Taking this into consideration, several natural products available at our lab, including flavonoids, were ranked by lowest binding energy and by the molecular interaction formed with the E6 protein on the E6AP binding site, as summarized in Table 1. Figure 1 displays, in more detail, the atomic interactions of the best-scored molecules with a binding energy lower than −5.50 Kcal/mol.

The lower the score, the higher the predicted affinity between E6 and the natural product. Taking this premise into account, and evaluating the molecular docking analysis, two hydroxylated anthraquinones (alizarin and lucidin) and a flavanonol (taxifolin) appeared to be potential HPV16 E6 protein inhibitors, considering that their binding energies were lower than −5.50 Kcal/mol and each product seems to dock in the E6AP binding site. Zanier and colleagues revealed that the residues Cys51, Tyr70, Leu67, Gln107, Leu50, Tyr32, Val31, Val62, Arg55, and Arg131 are crucial for E6/E6AP interaction [9]; therefore, considering Figure 1, it is possible to see that the three compounds might form important interactions with the E6 protein. In addition, these authors demonstrated that a point mutation in the residue Leu50 decreases the ability of an LxxLL peptide binding and, as a result, the reduction of p53 degradation [9]. Coincidentally, the three identified phenolic derivatives are predicted to interact with Leu50 of the E6 protein (Figure 1), which might be a promising indication of E6 activity blockage. 

Since there are no co-crystallized structures (E6 protein and ligands) available to validate the molecular docking results, we decided to use molecular dynamics simulations to perform binding mode studies and to predict the stability of the docked complexes. To assess the dynamic behavior of the complexes, the time-dependent root-mean-square deviation (RMSD) of all protein atoms was calculated using the original complexes as reference. A higher RMSD (>1 nm) is indicative of a molecule with an unstable nature [28]. The E6/alizarin complex showed RMSD between 0.3 and 1.8 nm as well as prominent conformational changes throughout the trajectory. RMSD increases until 170 ns, and thereafter, becomes constant at ~1.3 nm (Figure 2A). The higher RMSD is because E6 undergoes a huge conformational change upon binding the alizarin. Earlier studies have shown that E6 is a very flexible protein (RMSD 0.5 nm) [29]. Trajectory analysis suggests that the N- and C-domain moves away from ~3.5 to ~60 Å (Figure 2C), and the p53 binding pocket opens up widely. However, the structure of the N- and C-domain remains unchanged. Alizarin seemed to bind near the loop (aa 77–90), which is the interface of E6 and E6AP, and potentially prevents the binding of E6AP (Appendix A). 

In contrast, the binding of lucidin to E6 does not induce many conformational changes. The RMSD profile (0.4–0.8 nm) of E6/lucidin complex is similar to the apo E6. Its RMSD is ~0.4 nm until ~80 ns, and after that, it increases to ~0.8 nm throughout the trajectory. Trajectory studies suggest that lucidin could bind to E6 at two sites. Site I is located at N-terminal. The lucidin binding cavity is lined up by hydrophobic residues. Y43 makes π-π stacking interactions with lucidin. I23 and F47 form hydrophobic contacts with the aromatic ring of lucidin. H24 forms strong electrostatic interactions with lucidin (Figure 2D). Superimposition on the crystal structure suggests that the lucidin binding site I conformation does not change. However, the superimposition of the ternary complex E6/E6AP/p53 structure (PDB ID: 4XR8) onto E6/lucidin suggests that lucidin at site I is at the interface of p53 and E6. Therefore, it might prevent the binding of p53 with E6 (Appendix A). 

Natural compounds, in particular flavonoids, are known to bind E6 at multiple sites [30]. Similarly, lucidin also interacts with E6 at site II, which is at C-domain. At site II, lucidin interacts with polar residues K68 and R135. These residues form a H-bond interaction with lucidin. The residue Y76 rearranges to form π-π stacking interactions with the lucidin molecule. The overall conformation of the C-domain of E6 remains the same. Y79 has moved back to accommodate lucidin and forms a hydrophobic interaction with it. Similarly, H126 and I128 mainly form hydrophobic contacts with lucidin (Figure 2E). The dynamic behavior of the individual residue was estimated by root-mean-square fluctuations (RMSF) calculations, which denotes the positions of the residues across the simulation trajectory. All residues exhibit very low fluctuations, except the terminal residues (Figure 2B). Particularly, lucidin-interacting residues have very low RMSD (0.2–3 nm). However, the lucidin interaction with E6 at site II did not overlap with E6AP or the p53 binding site. Nevertheless, it induces a conformational shift of helix 67–78 (Appendix A). 

The RMSD of the E6/taxifolin complex is slightly high. It increases to 1.8 nm initially, and later, stabilizes at ~1.3 nm. The initial increase in RMSD can be attributed to the partial unfolding of E6 upon taxifolin binding. The α helical fragment (63–78) connecting the N- and C-terminal domain unwinds during simulation. Similar to the E6/alizarin complex, E6 also adopts an extended conformation in which the N- and C-domains move apart (Figure 2F). Later, the N- and C-domains come close to each other. The taxifolin binds near the fragment (63–78) and remains there throughout the simulation. This corresponds to the binding site of E6AP, and therefore, taxifolin might interfere in the binding of E6AP to E6 (Appendix A). However, the binding of taxifolin also induces conformational changes in E6. 

Even though structure-based drug design tools can predict the binding modes and affinity of compounds to targets with considerable accuracy and efficiency, the top results must be validated by in vitro studies.

### 3.2. Alizarin, Taxifolin, and Lucidin Cytotoxicity Studies in Cervical Cancer Cells 

We decided to examine the action of these three phenolic compounds in HPV-18 positive HeLa cells, HPV-16 positive Caski cells, HPV-negative C33A cells, and NHEK (a non-carcinogenic/spontaneously immortalized human keratinocyte cell line). In the first screening assay, cells were exposed to the three products at concentrations of 10 µM and 100 µM for 24 and 48 h, and the results of this screening are shown in Figure 3.

As can be observed in the results for 24 and 48 h of contact, no significant cytotoxicity towards NHEK (non-carcinogenic cell line) or C33A (HPV-negative cell line) was observed for almost all compounds at the lowest concentration. According to Figure 3A, the viability of HPV-positive cell lines after 24 h contact is around 75% or higher for nearly all compounds, while it can be observed in Figure 3B (48 h contact) that alizarin, taxifolin, and lucidin increased the reduction in the viability of HPV-positive cell lines. Since all derivatives demonstrated a preferential cytotoxic effect towards HPV-positive cells at 48 h, concentration vs. response studies were performed at this exposition time, and the concentration required to reduce the cell viability by 50% was estimated. The results obtained are displayed in Figure 4, with the dose–response curves presented in Appendix A, and some of them should be highlighted.

Interestingly, the results presented in Figure 4, Appendix A, and Table 2 confirm the selectivity of these anthraquinones and flavonoids for the E6-expressing cells once the concentration capable of reducing viability by 50% in HPV-positive cell lines are, in general, lower than 100 µM and, for C33A and NHEK cells, above 100 µM. In particular, lucidin greatly affected the viability of both HPV-positive cell lines, requiring 23 µM and 45 µM to decrease the cell viability by 50% in HeLa and Caski, respectively. On the other hand, alizarin and taxifolin showed less cytotoxicity for HeLa (alizarin = 64 µM; taxifolin = 63 µM) and Caski cells. Nevertheless, all compounds seemed to have a higher effect in HeLa rather than in Caski cells, and for that reason, we decided to perform molecular docking and the molecular dynamics simulations of the E6 protein from HPV18 (denoted as 18E6) with the three phenolic compounds. The 18E6 protein shares a 52% sequence identity with 16E6. Despite the low sequence similarity, the structures of both E6 proteins are highly conserved, with an RMSD of only ~0.8 Å (Appendix A). Therefore, we have retrieved the HPV18 E6 protein structure from the PDB (ID: 6SJV) and performed the docking studies with alizarin, lucidin, and taxifolin. In general, all compounds displayed lower binding energies (alizarin = −6.66 kcal/mol; lucidin = −6.31 kcal/mol; taxifolin = −6.52 kcal/mol) when compared to the docking scores of the HPV16 E6 protein. Regarding the results of molecular dynamics simulations, the phenolic compounds were predicted to bind at multiple locations on 18E6, as happens with 16E6. The 18E6 is highly flexible and adopts an extended form during simulation. These outcomes suggest that alizarin, lucidin, and taxifolin are capable of binding to 18E6 and to 16E6. Considering the in silico studies and the cell viability results, we selected HeLa cells for further analyses. 

Microscopic observations on HeLa cells treated with the concentrations displayed in Table 2 of all anthraquinones and flavonoids were performed to evaluate their effects on cell morphology and to observe any characteristics of cell death, as presented in Figure 5. Luteolin was used as a positive control at 23 µM (the concentration required to diminish cell viability by 50%) in HeLa cells, as shown in Appendix A and previously calculated by Cherry and colleagues [10]. In the DMSO-treated cells, their original morphology was maintained throughout the 48 h, with only an evident increase in the confluence. On the contrary, treated cells revealed morphology changes with typical features of apoptosis induction, such as rounding, shrinkage, vacuolization, formation of echinoid spikes, and nuclear fragmentation of cells [31,32], in a time-dependent manner. Moreover, all compounds seemed to have reduced the growth rate of HeLa cells at 48 h of treatment when compared with the control. 

### 3.3. Evaluation of p53 and BAX Protein Levels in Anthraquinone and Flavonoid Treated Cells 

Afterward, we sought to determine if alizarin, lucidin, and taxifolin were able to interfere with E6 function and, consequently, protect the p53 tumor suppressor from degradation, resulting in the re-establishment of p53 levels in HeLa cells. Luteolin, a flavonoid already described in the literature as able to restore p53 levels and having activity in HPV-positive cells [10,33], was used as a positive control. Data represent the pooled results from three independent Western blots, performed with samples acquired in three independent in vitro studies with different cell passages, prepared from lysates collected during the experiments. Representative blots are presented in Figure 6. Complete Western blots are presented in Appendix A. Results presented in Figure 6A evidence that the treatment of HeLa cells with alizarin showed no significant effect in restoring p53 levels. Meanwhile, lucidin and taxifolin were able to significantly increase p53 protein levels when compared to the DMSO-treated cells, where almost no p53 expression was detected [34]. Considering that normalization makes quantitative analysis more accurate, we normalized the intensity of the p53 band by dividing it by the intensity of the β-actin (housekeeping) band for each sample [35,36]. The ratio of the p53 protein to β-actin was then used to compare the p53 protein abundance in the treated cells to the negative control. As expected and observed in the Western blot, luteolin led to a significant accumulation of p53 protein levels. To demonstrate that lucidin and taxifolin only increase p53 protein levels without affecting p53 mRNA transcripts, we performed an RT-PCR. The treated cells showed no effect on p53 mRNA levels when compared to DMSO-treated cells (Appendix A), which suggests that lucidin and taxifolin have no influence on nuclear machinery and only affect protein levels, indicating p53 protein protection from E6-mediated degradation. Thus, to confirm that these compounds inhibit the E6-dependent degradation of p53, the p53 protein level was assessed on HPV-negative cell lines (C33A and NHEK, Appendix A). Complete Western blots are presented in Appendix A. The Western blot showed that p53 levels did not change upon treatment of C33A and NHEK cells with alizarin, taxifolin, and lucidin, corroborating that these compounds are specific for E6-expressing cells. However, to confirm the role of lucidin and taxifolin on HPV-positive Hela cells, some additional studies were considered, such as the control of BAX levels by Western blot, the evaluation of apoptosis induction by chromatin condensation and caspase 3/7 activity assays, and the study of cell cycle modulation by flow cytometry. 

In response to cell damage or induced stress, the p53 tumor suppressor functions as a transcription factor, facilitating the transcription of several genes implicated in cell cycle arrest or apoptosis, depending on the cellular environment, damage extension or other unknown factors [37]. Considering that lucidin and taxifolin caused a significant increase in p53 protein levels, we decided to evaluate if they were able to restore the p53 transcriptional activity. In line with this, we evaluated the levels of the BAX proapoptotic protein by Western blot. The BAX protein levels increased in HeLa cells treated with taxifolin and lucidin (Figure 6B) when compared to DMSO-treated cells. As expected, luteolin caused an increment in BAX levels [27] and alizarin did not exert any significant effect, which corroborates the results obtained for p53 levels. Altogether, these data demonstrate that lucidin and taxifolin can rescue p53 levels, which in turn increases the BAX pro-apoptotic protein levels in HPV18-positive cells.

### 3.4. Lucidin and Taxifolin Induce Apoptosis in HPV18-Positive Cells 

Since the increased level of the BAX protein suggests cell apoptosis induction by taxifolin and lucidin treatment, we decided to evaluate this phenomenon in higher detail. Apoptosis is a complex physiological process mediated by a succession of effectors and regulators and occurs via one of two major pathways: the extracellular pathway, which is triggered by the binding of a ligand to a death receptor, and the intracellular pathway, which is triggered by mitochondrial events following stress signals from within the cell [18]. Both pathways have a common feature which is the activation of executioner caspases (3, 6, 7) [18,38]. Besides that, these pathways lead to characteristic morphological alterations such as cell shrinkage and blebbing, the formation of apoptotic bodies, chromatin condensation, DNA fragmentation, the breakdown of the nuclear membrane, and increase in mitochondrial membrane permeability [39]. Therefore, we decided to evaluate chromatin condensation [25,40] and also caspase 3/7 activity [40], as presented in Figure 7. 

The nuclear morphological changes which occurred after treatment with these compounds were assessed by double staining with Hoechst 33,342 and PI. Hoechst 33,342 is commonly used to identify chromatin condensation and fragmentation by blue-staining the nuclei of apoptotic cells, while the PI (red color) stains cells that have lost membrane integrity [25]. The negative control (DMSO-treated cells) mostly shows a low blue color. After treatment with these phenolic compounds, some cells displayed a bright blue-colored nucleus (chromatin condensation), indicated by white arrows in Figure 7A, which is indicative of apoptosis. Necrotic cells, indicated by the orange arrows, were also observed with a higher incidence in cells treated with alizarin (Figure 7A). The percentage of apoptotic and necrotic cells was calculated, and the results are presented in Table 3. The treatment with taxifolin, lucidin, and luteolin led to a higher percentage of apoptotic cells when compared to DMSO-treated cells, which is in agreement with the BAX results. HeLa cells treated with alizarin displayed a slight increase in the percentage of apoptotic cells, which could be via a p53-independent pathway and a significant increase in the percentage of necrotic cells. 

Caspase 3/7 activity was evaluated with a commercial kit, and a known apoptosis-inducing agent, staurosporine, was used as a positive control. Figure 7B shows that lucidin and taxifolin treatment significantly increased caspase 3/7 activity, similar to the positive control. Once again, alizarin showed no significant effect on caspase 3/7 activation, while luteolin slightly induced caspase 3/7 activity. Another hallmark of apoptosis is DNA fragmentation, and therefore, we evaluated the sub-G1 population upon HeLa treatment with all phenolic compounds for 48 h and 72 h. Upon treatment for 48 h with lucidin and taxifolin, the percentage of sub-G1 cells was not significantly different from the DMSO-treated cells, but after 72 h, both compounds led to a significant increase in the sub-G1 population (Appendix A). One plausible explanation for this effect, considering that both compounds increase BAX expression and activate caspase 3/7, could be the fact that fragmented DNA is one of the last events of apoptosis (late apoptosis). The alizarin and luteolin treatment caused a substantial increase in the sub-G1 population at both 48 h and 72 h (Appendix A), which was expected for luteolin but not for the alizarin treatment. Bearing in mind that alizarin treatment has no effect on BAX protein levels, nor activates caspase 3/7, the increase in the sub-G1 population might happen via a caspase-independent cell death mechanism. In addition, we also performed an evaluation of cell cycle distribution to understand if taxifolin and lucidin were able to induce cell cycle arrest. For this, HeLa cells were treated for 48 h, and the cell cycle was assessed by PI staining followed by a flow cytometry analysis. We did not observe significant changes in cell cycle phase distribution of lucidin- and taxifolin-treated cells—the most promising compounds throughout this work—in comparison to DMSO-treated cells. Nevertheless, the luteolin and alizarin treatment increased the cell population in the S phase, and therefore, we assessed the p21 protein expression in HeLa cells to confirm the results obtained by flow cytometry. Complete Western blot is presented in Appendix A. The results shown in Appendix A are in agreement with the cell cycle analysis, since it is possible to observe that only the alizarin and luteolin treatment led to an increase in p21 levels when compared to the DMSO-treated cells. This behavior could indicate that the p21 is elevated through a p53-independent pathway, as alizarin almost had no effect on p53 or BAX protein expression. Consequently, these results suggest that alizarin could regulate the cell cycle via a p53-independent pathway. Similar results were reported by Zhu and colleagues, where a synthetic compound was able to induce S-phase arrest and increase p21 levels through a p53-independent mechanism [41].

## 4. Discussion

Cervical cancer is expected to continue to be a major burden in developing countries despite the existence of prophylactic vaccines. This is due to the lack of screening programs, limited resources, and access to health care, as well as the fact that vaccines are not able to exert a therapeutic effect on an established infection. Current treatments consist of surgery, radiotherapy, and chemotherapy, which cause infertility, several side effects to patients, and are expensive [7,36]. Considering that the p53 function is abrogated by the E6 protein in HPV-induced cancers, one line of investigation to pursue is the discovery of anticancer agents that could block the E6 protein’s role [17,42]. According to the literature, the focus has been on the inhibition of the E6/E6AP interaction, and a recently published work explored the direct interaction of E6/p53 proteins [12,17]. 

Thus, in the present study, we report the use of in silico tools to find E6/E6AP/p53 complex formation inhibitors based on the great potential demonstrated by flavonoids against cancer [18]. By molecular docking, we have identified three compounds (alizarin, taxifolin, and lucidin) able to adjust in the druggable binding pocket of E6 (Figure 1 and Table 1). In particular, these phenolic derivatives established important contacts with key amino acid residues for the E6/E6AP interaction [9,11]. According to Zanier and colleagues, compounds with a high affinity toward the E6AP binding pocket are likely to cause the inhibition of E6 in HPV-positive cells [11]. The MD simulations suggest that lucidin can block the binding of E6AP and p53 to the E6 protein of HPV16 and 18, while alizarin and taxifolin can impair the E6/E6AP binding with some conformational change in E6 proteins. Thereafter, we decided to verify if these three compounds could affect the viability of HPV-positive cells without affecting HPV-negative and non-tumoral cells. 

First, a screening assay based on two different concentrations (10 µM and 100 µM) of these anthraquinones and flavonoids was performed using luteolin as a positive control and DMSO-treated cells as a negative control (Figure 3). Luteolin was already described as being able to block E6 protein function, increase p53 levels, and decrease cell viability in HPV-positive cells; however, this flavonoid is also able to display off-target toxicity in HPV-negative cells. This issue reduced the interest in luteolin as a potential therapeutic intervention in HPV infection and cervical cancer [10]. Therefore, it is important to find compounds that mainly affect the viability of HPV-positive cells. Lucidin, taxifolin, and alizarin seemed to exert a selective cytotoxic effect on HPV-positive cells without significantly decreasing the viability of HPV-negative or non-tumoral cells. Consequently, we have calculated the concentration required to affect cell viability by 50% of each compound (Figure 4 and Appendix A) and found that the values are in the same range of other works describing compounds as able to inhibit E6 protein action in HPV-positive cells [10,12,36]. Although alizarin was computationally predicted to be the strongest inhibitor to be tested (highest binding energy), it presents the highest concentration that causes cell viability reduction by 50% in HPV-positive cells when compared to lucidin and taxifolin. Moreover, at a molecular level, we found that only lucidin and taxifolin were able to rescue p53 from E6-mediated degradation in HPV-positive HeLa cells (Figure 6A) without affecting the p53 levels on C33A or NHEK cells (Appendix A). This suggests that these compounds can interfere in the E6 recognition by p53, as was predicted in the in silico studies. The different behavior of these three compounds in HPV-positive cells could be explained by the MD simulations since they predicted that lucidin forms a highly stable complex with E6 protein of HPV16/18, and thus, it is more likely to impair the formation of the E6/E6AP/p53 complex. Although molecular dynamics data make us speculate that alizarin and taxifolin form a less stable complex with the E6 protein of HPV16/18, the aa residues involved in their binding to E6 are not the same, and taxifolin interacts strongly with the E6 protein, which could be the reason for their different actions in HPV-positive HeLa cells. Moreover, the molecular data are in agreement with previous works that describe other compounds as able to rescue p53 protein levels in HeLa cells [12,43]. 

In response to cellular stress, p53 can either keep cells alive (cell cycle arrest, senescence, and autophagy) or kill the cells (apoptosis and ferroptosis), depending on the context [44]. Therefore, we investigated if the compounds were able to activate the p53-dependent target gene by assessing BAX protein levels (Figure 6B) on HeLa cells. Taxifolin and lucidin were able to induce BAX protein activity in resemblance to the p53 protein results, which suggested apoptosis induction on HeLa cells. In addition, lucidin and taxifolin did not affect the BAX protein levels in the HPV-negative cell line (Appendix A), which seems to suggest that these compounds inhibit the E6-dependent degradation of p53. Apoptosis is responsible for maintaining normal cellular function, where the two most important proteins involved are caspases and the B-cell lymphoma 2 family. Caspases can be classified, according to their role, into initiator caspases (2, 8, 9 and 10) and executioner caspases (3, 6 and 7), which trigger apoptosis by two main pathways (extrinsic and intrinsic) [45]. Considering that treated cells display chromatin condensation with a significant percentage of apoptotic cells and an increase in active caspase 3/7 and BAX protein activity, it is very likely that the lucidin and taxifolin induce apoptosis by the intrinsic pathway. In fact, despite chromatin condensation and caspase 3/7 activity being common to both apoptotic pathways, BAX activity is mainly related to the intrinsic pathway [38]. 

As mentioned above, p53 is also able to induce cell cycle arrest, so we performed a cell cycle analysis by flow cytometry. Neither lucidin nor taxifolin-treated cells displayed significant alterations when compared to DMSO-treated cells. One probable explanation is the fact that the oncogenic potential of HPV-positive cells is driven by two oncoproteins, E6 and E7 [42]. Since we are only blocking E6 protein function, it is possible that E7 maintains its normal function. E7 can be associate with the retinoblastoma protein (pRb), which leads to its degradation and cell cycle progression. In addition, E7 binds and degrades p105 and p107, which control cell cycle entry and cause epigenetic reprogramming of the cell, which is important for stimulation of cell cycle entry and progression [4]. Taken together, we found that lucidin and taxifolin were able to decrease HPV-positive cell viability, restore p53 protein levels, and induce apoptosis in HPV-positive HeLa cells. Furthermore, taxifolin has been already identified as a potential antineoplastic agent in several cancers such as breast [46], colon [47], lung [48], osteosarcoma [49], and prostate cancer [50]. Usually, tumors express mutant or wild-type p53. Regarding the tumors expressing wild-type p53, promising approaches to treat cancers could be to block major p53 inhibitors, Mdm2 and MdmX, or viral E6 oncogene in HPV-driven cervical cancers [51]. There have been several studies published that describe the use of compounds to restore p53 function and activity. For example, molecules as nutlins, spiro-oxindole compounds, and benzodiazepinediones have been described to induce p53 rescue by different pathways, in which the MDM2 inhibition is the mainly studied [52]. Therefore, lucidin and taxifolin could be a starting point for the design and development of compounds as an antiviral therapy for HPV infection and cervical cancer.

## 5. Conclusions

In this study, we demonstrated that in silico tools were helpful for screening several compounds and selecting three hits that considered the E6 protein as a target. In addition, two of the selected compounds, taxifolin and lucidin, revealed selective cytotoxicity for E6-expressing cells, rescued p53 protein levels, and induced apoptosis in HPV-positive HeLa cells. In the near future, it would be interesting to investigate the survival of HPV-positive stem cells and study the combinatory effect between lucidin and taxifolin to understand if their action could be potentiated. Additionally, the study of the analogs of these phenolic compounds would be valuable to find compounds with higher potency and to obtain relevant information about the structure–activity relationship.

## Figures and Tables

**Figure 1 cancers-14-02834-f001:**
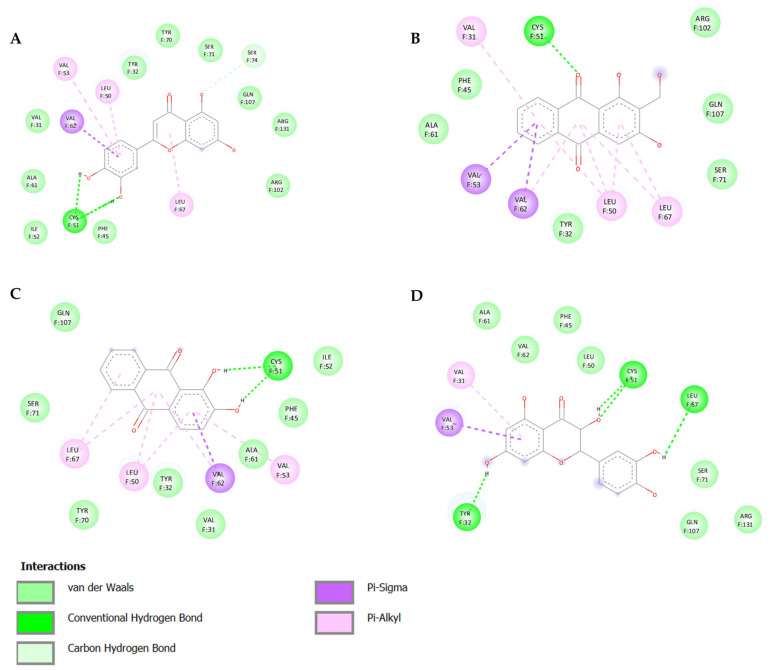
Binding modes and interactions of luteolin (**A**), lucidin (**B**), alizarin (**C**), and taxifolin (**D**) with the HPV16 E6 protein.

**Figure 2 cancers-14-02834-f002:**
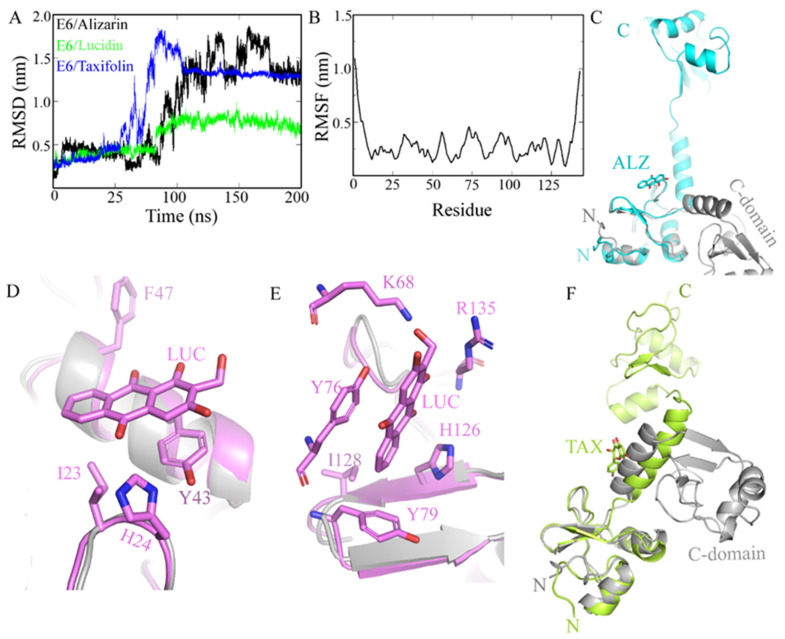
MD simulation study of E6/alizarin/lucidin/taxifolin complexes. (**A**) Root-mean-square deviation (RMSD) of E6 complexes. E6/lucidin complex is the most stable complex. (**B**) Root-mean-square fluctuations (RMSF) of E6/lucidin complex. (**C**) Trajectory analysis of E6/alizarin complex suggests that both N- and C-domain (cyan) move away from the initial structure (grey). Lucidin binds E6 at two sites. (**D**) Site I at N-domain; lucidin interacting residues are shown. (**E**) Site II at C- domain; lucidin residues that interact shown as stick representation. Simulated structure is shown in violet whereas the original structure is in grey. (**F**) During simulation, E6 adopts an extended conformation. Simulated structure is shown in lime-green whereas the original structure is in grey. Abbreviation- ALZ (alizarin), LUC (lucidin), TAX (taxifolin).

**Figure 3 cancers-14-02834-f003:**
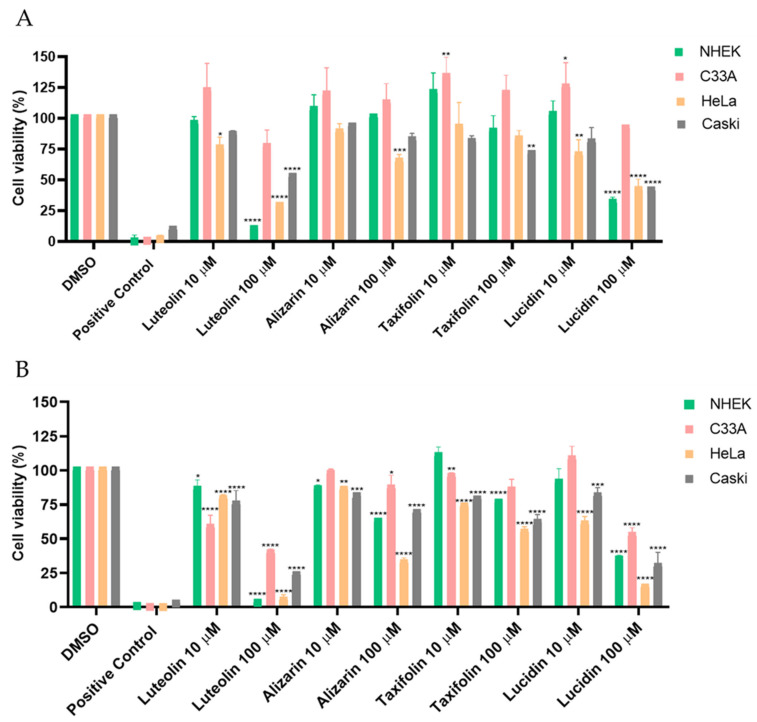
Cellular viability of different cell lines at 24 and 48 h incubation with phenolic compounds. All products were tested at 10 µM and 100 µM in HPV-positive (Caski and HeLa), negative (C33A) and non-carcinogenic (NHEK) cell lines by the MTT assay after 24 h (**A**) or 48 h (**B**) of treatment. Positive control corresponds to ethanol-treated cells. Percent viability was determined for each sample relative to the DMSO treated control samples. Data are presented as mean ± SD for three independent experiments (*n* = 3) with four technical replicates and analyzed by one-way ANOVA with the Bonferroni test. Significance was determined as *p*-values * < 0.05, ** < 0.01, *** < 0.001, **** < 0.0001.

**Figure 4 cancers-14-02834-f004:**
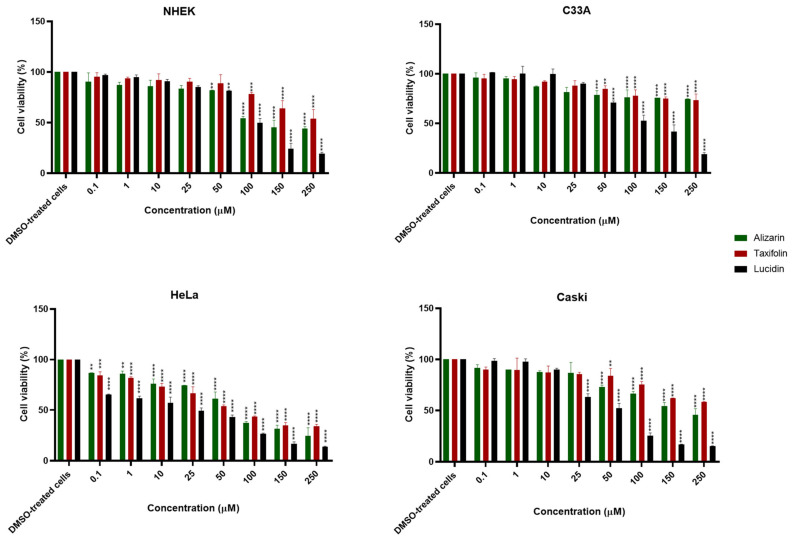
Effect of phenolic compounds on the viability of HPV-positive and negative cell lines. Cells were treated at several concentrations (0.01, 0.1, 1, 10, 50, 100, 150, and 250 µM) for 48 h, and cell viability was determined by the MTT assay. Data are presented as mean ± SD for three independent experiments (*n* = 3) with four technical replicates and analyzed by one-way ANOVA with the Bonferroni test. Significance was determined as *p*-values ** < 0.01, *** < 0.001, **** < 0.0001.

**Figure 5 cancers-14-02834-f005:**
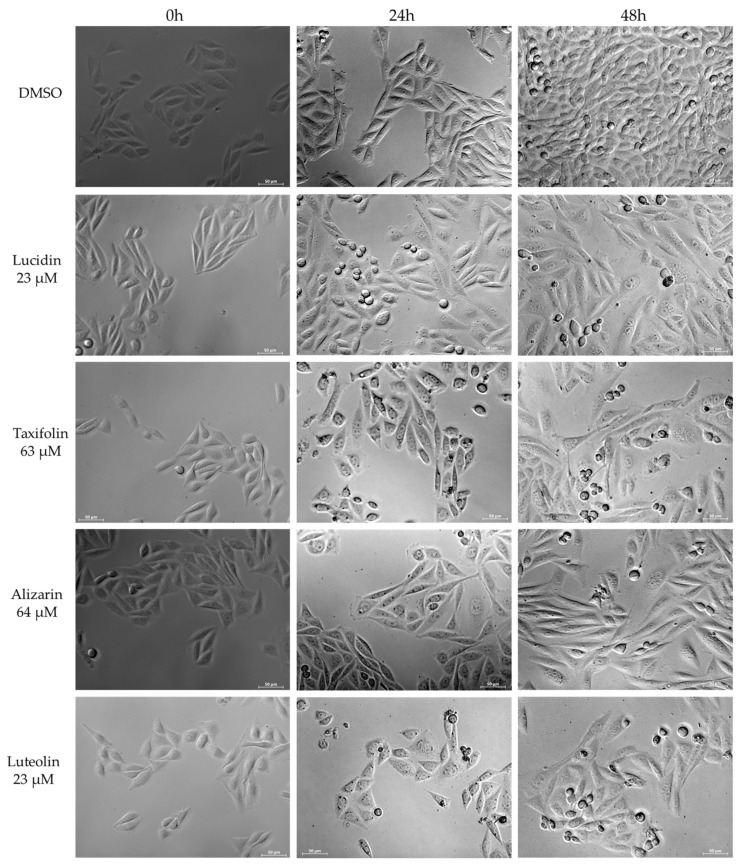
Phase contrast images showing HeLa cell morphology at 0, 24, and 48 h treatment at 20× magnification. Scale-bar: 50 µm.

**Figure 6 cancers-14-02834-f006:**
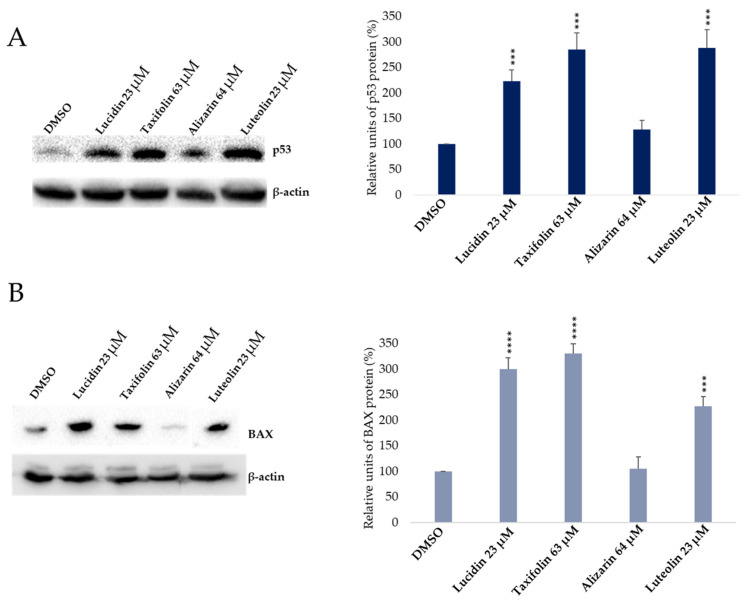
The effect of anthraquinones and flavonoids on p53 (**A**) and BAX (**B**) protein levels in HPV-positive Hela cells was evaluated by Western blot analysis after 48 h of treatment. Data were normalized against the β-actin and plotted as a percentage related to the DMSO-treated cells. Each data set represents the mean ± SD for three independent experiments (*n* = 3) performed with samples acquired in three independent in vitro studies analyzed by the *t*-Student test. Significance was determined as *p*-values *** < 0.001, **** < 0.0001.

**Figure 7 cancers-14-02834-f007:**
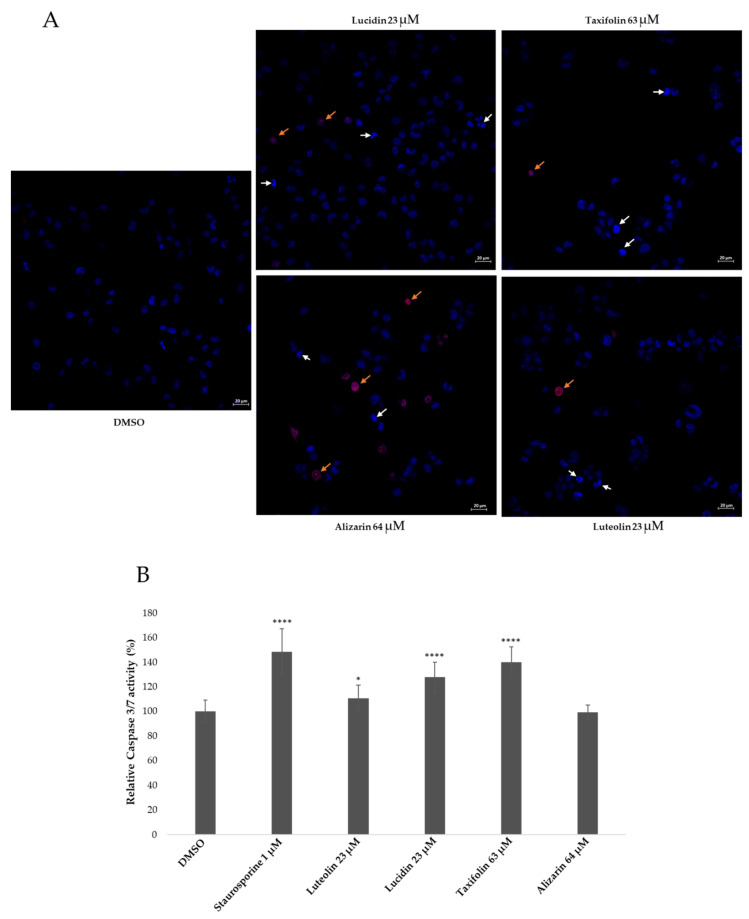
Effect of anthraquinones and flavonoids on nuclear morphological changes of HeLa cells at 48 h treatment (**A**). Cells were stained with Hoechst 33,342 and PI and observed with a fluorescence microscope under a magnification of 40×. Effect of anthraquinones and flavonoids on the caspase activity of HeLa cells, determined after 48 h of contact by the Caspase 3/7-Glo assay (**B**). DMSO-treated cells were used as a negative control, while incubation with 1 μM of staurosporine for 48 h was used as a positive control. Data are presented as mean ± SD, *n* = 3 (three independent experiments) and analyzed by the *t*-Student test. Significance was determined as *p*-values * < 0.05, **** < 0.0001.

**Table 1 cancers-14-02834-t001:** List of natural products, including flavonoids, binding energies, and main interactions with the HPV16 E6 protein on the E6AP binding site.

Natural Products	Structure	Binding Energy (Kcal/mol)	Main Interactions
Alizarin	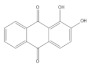	−6.52	Cys51, Tyr70, Leu67, Gln107, Leu50, Tyr32, Val31, Val62
Caffeic acid	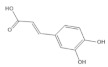	−5.42	Cys51, Tyr70, Leu67, Gln107, Leu50, Tyr32, Val31, Val62
Cyanidin-3-*O*-glycoside	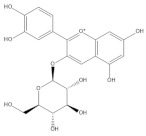	−5.45	Cys51, Tyr70, Leu67, Gln107, Leu50, Tyr32, Val31, Arg55
Cineol	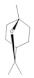	−4.73	Leu67, Tyr32, Cys51, Leu50, Val62
Kuromanin	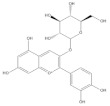	−5.22	Cys51, Leu67, Leu50, Tyr32, Val31, Val62
Ellagic acid	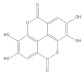	−5.36	Tyr70, Leu67, Gln107, Leu50, Tyr32, Val31, Val62, Arg131
Ferulic acid	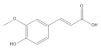	−5.45	Cys51, Leu67, Leu50, Tyr32, Val31, Val62
Gallic acid	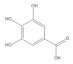	−4.43	Cys51, Tyr70, Leu67, Leu50, Tyr32, Val62
Genistin	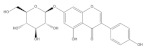	−5.29	Cys51, Tyr70, Leu67, Leu50, Tyr32, Val31, Val62
Linalool	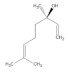	−4.56	Cys51, Tyr70, Leu67, Gln107, Leu50, Tyr32, Val31, Val62
Lucidin	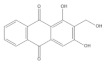	−5.87	Cys51, Leu67, Gln107, Leu50, Tyr32, Val31, Val62
*p*-coumaric acid	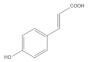	−5.43	Cys51, Tyr70, Leu67, Leu50, Tyr32, Val31, Val62
Quercetin-3-4′-di-*O*-glycoside	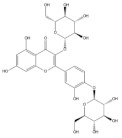	−3.66	Cys51, Tyr70, Leu67, Gln107, Leu50, Tyr32, Arg55, Val62, Arg131
Rosmarinic acid	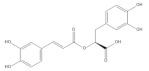	−5.11	Cys51, Leu67, Gln107, Leu50, Tyr32, Val31, Val62
Sabinene	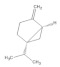	−4.53	Cys51, Leu67, Leu50, Tyr32, Val31, Val62
Syringic acid	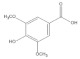	−4.69	Cys51, Leu67, Leu50, Tyr32, Val31, Val62
Taxifolin	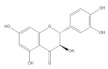	−5.63	Cys51, Leu67, Gln107, Leu50, Tyr32, Val31, Val62, Arg131
Rutin	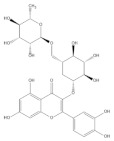	−4.76	Cys51, Tyr70, Leu67, Gln107, Leu50, Tyr32, Val31, Val62, Arg55, Arg131
Vanillic acid	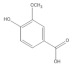	−4.44	Cys51, Leu67, Leu50, Tyr32, Val31, Val62
Luteolin	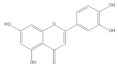	−6.28	Cys51, Tyr70, Leu67, Gln107, Leu50, Arg131, Tyr32, Val31, Val62

**Table 2 cancers-14-02834-t002:** Compound concentration required to reduce cell viability by 50% of three independent experiments (*n* = 3) determined by regression analysis.

Compound Concentration that Reduces Cell Viability by 50%				
Phenolic Compounds	NHEK	C33A	HeLa	Caski
Alizarin	>100 µM	>100 µM	64 µM	>100 µM
Lucidin	>100 µM	>100 µM	23 µM	45 µM
Taxifolin	>100 µM	>100 µM	63 µM	>100 µM

**Table 3 cancers-14-02834-t003:** Percentage of necrotic and apoptotic cells (%) with data presented as mean ± SD, *n* = 6. The analysis was performed by the *t*-Student test (statistical significance in comparison to DMSO-treated cells as *p*-values * < 0.05, ** < 0.01, *** < 0.001, **** < 0.0001).

	Percentage of Necrotic Cells (%)	Percentage of Apoptotic Cells (%)
DMSO	0.74 ± 0.6	1.43 ± 0.44
Lucidin 23 µM	1.78 ± 0.69 (*)	5.16 ± 0.69 (****)
Taxifolin 63 µM	2.54 ± 0.51 (**)	6.43 ± 0.44 (****)
Alizarin 64 µM	8.94 ± 0.35 (****)	3.24 ± 0.80 (**)
Luteolin 23 µM	2.56 ± 0.62 (*)	4.46 ± 0.91 (***)

## Data Availability

Data is maintained in this paper. Data is not publicly available due to privacy.

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
