# Peer review of "Taxifolin and Lucidin as Potential E6 Protein Inhibitors: p53 Function Re-Establishment and Apoptosis Induction in Cervical Cancer Cells"

_cancers, 2022, doi:10.3390/cancers14122834_

Round 1
Reviewer 1 Report
In revised version of manuscript Diana Gomes and co-authors addressed the Reviewer’ concerns. The experiment with C33A cells as a HPV negative cell line was performed (supp material). Only one repeat is presented but it looks convincing. The experimental repeats showing restoration of p53 protein and RT-PCR showing p53 mRNA were added to supplementary data.
The Authors monitored the level of p21 to confirm lack of changes in cell cycle (the Reviewer’ suggestion). Interestingly, it turned out that under luteolin and alizarin treatment, the level of p21 protein is elevated particularly, in the presence of alizarin. Whereas the level of p53 is little restored under alizarin treatment. It is almost comparable to DMSO treatment and as a consequence no BAX elevation is observed (Fig. 6). It could indicate the p21 is elevated through p53-independent pathway. I suggest adding Figure 1 to supplementary data with a short comment of possible regulation of cell cycle by alizarin via p53-independent pathway in a main text. It is worth presenting such observation.
Author Response
Reviewer 1:
#1 In revised version of manuscript Diana Gomes and co-authors addressed the Reviewer’ concerns. The experiment with C33A cells as a HPV negative cell line was performed (supp material). Only one repeat is presented but it looks convincing. The experimental repeats showing restoration of p53 protein and RT-PCR showing p53 mRNA were added to supplementary data.
Response 1: We thank the reviewer for recognizing the efforts we made to meet the requirements pointed out previously.
#2 The Authors monitored the level of p21 to confirm lack of changes in cell cycle (the Reviewer’ suggestion). Interestingly, it turned out that under luteolin and alizarin treatment, the level of p21 protein is elevated particularly, in the presence of alizarin. Whereas the level of p53 is little restored under alizarin treatment. It is almost comparable to DMSO treatment and as a consequence no BAX elevation is observed (Fig. 6). It could indicate the p21 is elevated through p53-independent pathway. I suggest adding Figure 1 to supplementary data with a short comment of possible regulation of cell cycle by alizarin via p53-independent pathway in a main text. It is worth presenting such observation.
Response 2: We agree with the revisor’s suggestion and the results of p21 protein expression were added to the manuscript and supplementary data, please see pages 21 and 5, respectively.
Reviewer 2 Report
I was Reviewer 2 on the prior version and submitted extensive comments. The majority of these have been addressed and overall the manuscript is improved.
- My primary concern was that the increases in p53 the authors report could be E6 independent and they needed to test HPV E6 negative cells. They noted in line 603 that this off target effect was reported with luteolin). This concern remains. Unfortunately, they chose to perform experiments with C33A cells and found that the lead compounds did not increase levels of p53 protein. The problem is that these cells express high levels of the stable mutant p53 (Arg 273 to Cys) as apparent in Figure S6. Furthermore, these cells express dominant negative p53 and less susceptible to apoptosis inducing chemicals (this is apparent in Figure S2) . The authors used NHEK cells in the viability assays and could have performed the p53 blots and apoptosis assays in these cells. There are other cell lines that express wild type p53. Without the proper cell control, their conclusion about specific E6 inhibition by these compounds could be wrong.
- Alizarin did not increase p53 protein but the authors dismissed its effect on apoptosis as a ‘slight increase’ (line 562) compared to the DMSO control. Their rationalization of this discrepancy is not convincing.
- The authors also responded that flow cytometry did not reveal a subgenomic, apoptotic population following treatment with these compounds. That’s a problem too – they suggested it was a timing issue but I don’t buy that.
- I previously noted that the text stated on several occasions that the compounds bound or complexed to E6, for which the authors have no biochemical data, and present only in silico modeling. Their letter states they agreed and removed such comments, but these still appear, e.g. Line 622 “although alizarin taxofolin form a less stable complex”. Check again carefully.
Other suggestions
- Figure S1 is difficult to interpret. It’s not at all clear how these compounds would block E6AP binding in the ribbon cartoons. Maybe a few E6 amino acid #s would help. The authors docked these compounds onto the E6/E6AP/p53 trimer. Why not E6 alone?
- Figure S2. Why is HeLa viability decreased by ~20% at submicromolar Lucidin?
- Figure S3 doesn’t add much to this paper. Would eliminate.
Author Response
The authors would like to acknowledge the careful revision and pertinent reviewer’s comments. All the questions were answered in the attached document and the recommended modifications were made, being properly highlighted at yellow in the revised manuscript file. With this, we believe that we have further clarified the addressed comments and improve the manuscript.

Reviewer 3 Report
While the authors made significant changes to the manuscript, including additional data analysis and description, the writing quality of the manuscript both in English language and scientifically is too low to approve publication. I recommend engaging a professional scientific writer to increase the quality to the expected standard.
Author Response
Reviewer 3:
#1 While the authors made significant changes to the manuscript, including additional data analysis and description, the writing quality of the manuscript both in English language and scientifically is too low to approve publication. I recommend engaging a professional scientific writer to increase the quality to the expected standard.
Response 1: We acknowledge the reviewer comment and have tried to increase the quality of scientific writing and English language throughout the manuscript.
Round 2
Reviewer 2 Report
no more comments
Reviewer 3 Report
No more changes required.
This manuscript is a resubmission of an earlier submission. The following is a list of the peer review reports and author responses from that submission.
Round 1
Reviewer 1 Report
The manuscript by Diana Gomes and co-authors addresses the potential of phenolic compounds to bind to E6 protein and as a consequence to inhibit the E6-mediated p53 degradation and induce apoptosis in HPV positive cells. The manuscript is clear, from in silico screening to in vitro tests. The manuscript also presents a table with the list of natural products, including flavonoids and their characteristics, additionally binding models for selected chemicals.
Despite experimental work seems to be adequate there are some concerns.
Authors decided to apply only one cell culture (HeLa) for further in vitro analysis. It should be also used C33A cells (HPV negative cell line) to show that observed effect of p53 rescue is specific. It is important to show that proposed chemicals act in specific way. There is possibility that p53 protein level would be increased as a general response to phenolic compounds. One type of experiment, applying C33A cells, as presented in Figure 6A,B is highly suggested.
Authors claim that p53 protein level is restored after selected chemical treatments. As I understand correctly, negative control is DMSO-only treated, information should be added in figure description. The p53 protein level seems to be high elevated compared to a negative control (Fig 6A), however there are no experimental repeats in supplementary material. All repeats (Western Blots) for p53 an BAX should be presented in supp. material.
Data concerning p53 mRNA levels upon chemical treatment is not shown however in Material section description of experiment (RT-PCR) is present. I recommend to show the results in supp. material.
In line with p53 and Bax elevation an increased caspase 3/7 activity is observed. No repeats are shown in supp. material. Additionally, Authors analysed changes in cell cycle phases but no alterations are observed (data not shown). To confirm this observation is good to monitor the level of p21 protein as well.
Since the Authors present potential applying of phenolic compounds in p53-based treatment in cervical cancer it would be interesting to discuss data concerning treatments of other cancers based on p53 activation/restoration.
Reviewer 2 Report
This manuscript describes in silico modeling of HPV E6 binding chemicals, which is interesting. While these predictive algorithms are powerful, the authors recognized that experimental back up is necessary. However, these data are a major weakness and detract from the significance of their findings. Careful and rigorous experiments are required.
The authors describe an in silico approach to identify novel compounds that inhibit HPV E6 function. Although the in silico screening was done with HPV-16 E6, the identified compounds show greater efficacy against HPV 18 E6 cells. The authors need to comment on the structural similarities and differences between 16 E6 and 18 E6 and how the docking experiments compare with the cellular experiments.
There are multiple technical flaws. The viability assays need to be repeated to include three independent experiments. The curve fitting needs to be revisited as an IC50 cannot be calculated if 50% of cells are not dead even at the highest concentration. The effects of the identified compounds on p53 and Bax expression need to be repeated in a HPV negative cell line to draw any conclusions. Throughout the manuscript the wording of the observed in silico binding modes needs to clearly state that it is hypothetical binding of each compound to HPV-16 E6. At places in the manuscript as seems to imply these observations are experimentally proven.
The entire manuscripts needs to be proof-read and corrected.
Specific comments.
- Line 68:Which E6 is exactly 160 amino acids?
- Line 71: “E6AP a ..connecting bridge..” This is not correct. E6AP does not bind p53, rather binds to and allosterically alters E6 to enable binding to p53.
- Line 90: “have gained huge” ... what?
- Line 90-95 : what’s the specificity of natural compounds (for E6)?
- Line 292: “in the literature as originating a re-activation of the p53-mediated pathway.. “
- State that HPV 16 E6 is being used in the docking experiments.
- Table 1 is incomplete.All structures are not depicted.
- Line 299: How do these calculated binding energies compare to previously published docking experiments ?
- Line 318... 326: Should be toned down. These are in silico experiments that show how a compound might bind to E6. How about the higher energy binding modes that are predicted for each of those compounds. Are they predicted to interact with the same amino acids?
- The authors observe lower binding scores with different flavonoid derivatives than Luteolin. Can this be experimentally validated?
- Line 328: co-crystal of what? Presumably compound and E6- they should state that.
- Line 355: Does alizarin also bind at different sites on E6?
- Line 366: ...Is Lucidin binding at both site is required for its action? This is highly speculative. The authors should validate binding to HPV-16 E6 prior to making these claims.
- Figure 3: What’s the positive control and what is it concentration in A and B? Text says Luteolin but that is labelled separately. Is the statistical significance compared to the individual DMSO control? Statistics for comparison between cell lines at individual drug exposures need to be added.
- Line 440: ...”that alizarin, taxifolin and lucidin revealed higher cytotoxicity in HPV positive cell lines”... Statistical significance needs to be added to the graph comparing the cell lines. It’s unclear what they have done for these comparisons.
- Figure 4B: . An IC50 cannot be calculated if the highest concentration does not even reduce cell viability by more than 50%. e.g. Taxifolin causes about 60% toxicity and supposedly has an IC50 of 63 uM. How is this possible?
- Figure 4A needs to have the actual fitted curve instead of the connecting lines.Why do Caski, Hela, and NHEK have less than 50% viability with Lucidin but only Hela and Caski have an IC50? The data analysis needs to be revisited by a statistician. Additionally, the results and interpretations may benefit from additional higher concentration to accurately determine the IC50. Experiments should be performed three independent times. The authors should compare in silico modeling predictions with in vitro experimental data.
- Line 482: Data needs to be newly interpreted after increasing n numbers and revisiting the curve fitting calculations. An IC50 cannot be calculated if the drug does not even cause 50% reduction. Ideally cell viability should at minimum be below 20%.
- Line 489: “All compounds evidenced a higher effect in HeLa rather than in CaSki…” What’s the point of the docking experiments then, which presumably were performed with the HPV-16 E6 structure? Do they presume 16E6 is identical with 18E6? Can they explain why compounds were more effective in 18 E6 (HeLa) than HPV-16 E6 (CaSki)?
- Line 493 : What is the luteolin concentration? What’s the negative control?
- Figure 5:What do these data add other than what viability shows?
- Figure 6A: The Bar graph does not seem to reflect the western blot p53 inductions as they look more like two-fold changes.
- Figure 6A, B: Totally cropped lanes are unacceptable. The entire images should be shown.Comparing protein band intensity changes between groups of p53 vs Bax and their individual bar graphs vary widely. Bax bands are minimally different in the western image and p53s are very different. Yet Bar graphs show similar fold changes.
- Line 541 ... ‘caused an increment in p53 levels’... What does this mean?
- Figure s6 and 7: need to include a non-HPV cell line for comparison to Hela at the same concentrations of compounds tested.
- Figure 7B: They state cell cycle changes were observed using flow cytometry.Did they find a subgenomic population indicative of apoptotic cells?
- Recommend include an HPV16 cell line to Figures 6 and 8 to be able to draw comparisons of docking results to the in vitro results.
- Discussion:Specificity: need to state throughout manuscript that docking was performed with HPV 16 E6. HeLa express HPV-18 E6 . They should address structural differences between 18 and 16 otherwise docking is pointless.
- Line 647: Need to test in non-HPV cell to draw this conclusion..
- Line 648: Cannot be explained as HPV 16 E6 was used and effects are presented with 18 E6.....
Reviewer 3 Report
Gomes et al select several potential inhibitors of the HPV16 E6 protein based on docking studies and begin a preliminary characterization of the effects of these compounds on cell viability, p53 levels, apoptosis.
Major points.
1) The structure used is for HPV16 E6, yet studies are largely performed on HeLa cells, which express HPV18 E6. These proteins are less than 57% identical, so it is highly unlikely that the docking interactions made by the small molecules under study are identical between these two types of high risk HPV proteins. Modelling of HPV18 E6 needs to be done, and docking of the small molecules on that model tested. Will this yield data supportive of interaction and effects on HeLa cells? I suspect not. Show me the equivalent of figure 1 with HPV18 E6 to convince me that these structural prediction with HPV16 E6 is actually relevant.
2) Table 1 is not useful. Make it supplementary data at best.
3) Figure 4 needs a selectivity index calculated/estimated. The data provided are not sufficient to calculate this. Why is no luteolin data presented here?
4) Figures 5 and 7 need to be quantified in some way. As representative data, I did not find it convincing.
Although mostly well written, some of the English needs to be improved. Most times this is related to the choice of an inappropriate word or a missing word. Some examples from the first page alone include: line 25 the word existing should be represents; line 29 should read "allowed us to select"; line 33 should be "fourth leading cause"
Other minor issues:
The type of HPV E6 used for the computational biology approaches should be explicitly stated as HPV16 for the structure 4XR8 on lines 116 and 289.
As all HPV E6's are not close to being identical the statement on line 68 should focus on high risk HPV16 specifically.
The HPV status of the cell lines on line 166 should be stated as HPV18 and 16 respectively.
Bioinformatics does not really including docking studies (line 618). That is computational biology. Docking models are also predictions as they are far from perfect, yet the manuscript treats these as fact (see line 319 etc). This has to be stated correctly instead of appearing to make the assumption that they are perfectly correct.
Most of the text for section 3.4 of the results can be deleted as it is unnecessary.
Reviewer 4 Report
The authors identified natural compounds predicted to interact with E6 and block the interaction with p53 using in silico prediction models. The authors verified potency of 3 agents in in vitro cell lines expressing E6, and two of these agents led to restoration of p53 levels and increased levels of cell death.
While this study is important and suggests new therapeutic targets for HPV+ malignant diseases, the manuscript in its current form lacks scientific language proficiency, and would significantly benefit from engaging a professional scientific writer/editor.
Fig. 3: What does the dotted line indicate? The authors should rethink the presentation of this data. It is not easy to distil the critical results from this data presentation.
Fig. 4: From which data was the IC50 calculated? Raw data should be included. If it was calculated based on the cell viability shown in A, then the IC50 of Lucidin on HeLa cells doesn’t appear to be 23uM. Statistics should be incorporated to Fig 4A (e.g. AUC). Since tables should not be part of figures, the authors might want to show the sigmoidal curves at B with IC50 indicated and instead have the Table as Table 2.
Fig. 5: Quantify cell death.
Fig. 6: Show three replicates in WBs